# Anti-Oxidative and Anti-Inflammatory Effect of Protaetia Brevitarsis-Derived Protein Hydrolysates in Adipose Tissues of Obese Mice

**DOI:** 10.3390/ijms262110352

**Published:** 2025-10-24

**Authors:** Jun-Koo Kang, Eun Hye Lee, Bo Hyun Yoon, Minji Jeon, Jae-Wook Chung, Phil Hyun Song, Tae Gyun Kwon, Yun-Sok Ha, So Young Chun, Syng-ook Lee, Bum Soo Kim

**Affiliations:** 1Department of Urology, School of Medicine, Kyungpook National University, Daegu 41566, Republic of Korea; kangjk0082@naver.com (J.-K.K.); jeus119@hanmail.net (J.-W.C.); tgkwon@knu.ac.kr (T.G.K.); yunsokha@gmail.com (Y.-S.H.); 2Joint Institute for Regenerative Medicine, Kyungpook National University, Daegu 41566, Republic of Korea; eun90hye@gmail.com (E.H.L.); bobo1904@naver.com (B.H.Y.); njiya120@naver.com (M.J.); soyachun99@naver.com (S.Y.C.); 3Department of Urology, College of Medicine, Yeungnam University, Daegu 42415, Republic of Korea; sph04@hanmail.net; 4Department of Food Science and Technology, Keimyung University, Daegu 42403, Republic of Korea

**Keywords:** obesity, anti-oxidative, anti-inflammatory, protaetia brevitarsis-derived protein hydrolysate, high fat diet

## Abstract

Obesity is a major global health issue linked to metabolic disorders, chronic inflammation, and systemic complications, with high-fat diets (HFDs) playing a key role by disrupting intestinal balance and promoting oxidative stress. This study investigates Protaetia brevitarsis-derived protein hydrolysate (PBPH), an insect-derived bioactive peptide extract, as a potential intervention to counteract HFD-induced metabolic disturbances. Female ICR mice were divided into three groups: control diet, HFD, and HFD + PBPH, with PBPH (obtained by alcalase hydrolysis and ultrafiltration) administered daily for eight weeks. Researchers assessed adipokine levels, inflammatory cytokines, antioxidant enzymes, and apoptotic markers using qPCR, ELISA, histology, and immunohistochemistry. PBPH supplementation significantly improved metabolic parameters by lowering leptin, adipsin, resistin, IL-1β, TNF-α, and IL-6, while restoring antioxidant balance and reducing pro-apoptotic signals. Histological analyses confirmed preserved intestinal tissue and reduced inflammation. Overall, this study highlights PBPH’s promising therapeutic role in addressing obesity-related metabolic dysfunctions through its multifaceted effects on inflammation, oxidative stress, and apoptosis. It underscores the potential of insect-derived peptides as sustainable, innovative dietary interventions for improving metabolic health.

## 1. Introduction

Obesity is a growing global health concern, with its prevalence increasing dramatically over the past few decades [1]. Changes in modern lifestyle, such as physical inactivity, sleep disturbances, and dietary habits have contributed to this increasing prevalence of obesity and related metabolic syndromes [2,3]. Obesity contributes to metabolic disorders such as type 2 diabetes, cardiovascular diseases, and chronic inflammation [4,5,6]. Among the various factors influencing obesity and its associated complications, diet plays a crucial role [7]. High-fat diets (HFDs) have been widely studied for their impact on metabolic dysfunction, oxidative stress, and inflammatory responses in various tissues, particularly in the gastrointestinal system [8,9,10]. The intestinal environment plays a crucial role in obesity and metabolism. HFD consumption not only leads to excessive weight gain but also triggers significant changes in intestinal homeostasis [10]. These alterations include disruption of the intestinal barrier function, changes in gut microbiota composition, and activation of inflammatory pathways [10]. The intestine, as the primary site of nutrient absorption and metabolism, is particularly susceptible to HFD-induced damage, which can lead to both local and systemic inflammation [11]. The imbalance between pro-inflammatory and anti-inflammatory signals induced by HFD consumption has been linked to increased adipokine secretion, oxidative damage, and apoptosis, ultimately leading to gut dysbiosis and metabolic dysfunction [12].

Recent research has highlighted the complex relationship between adipokines and intestinal inflammation in obesity [13,14]. Adipokines, including leptin, adipsin, resistin, and adiponectin, are bioactive molecules primarily secreted by adipose tissue that play essential roles in energy homeostasis, inflammation, and metabolism [15,16]. In obesity, the dysregulation of these adipokines contributes to a chronic low-grade inflammatory state, which can exacerbate intestinal dysfunction and metabolic disorders [17,18]. The inflammatory response triggered by HFD consumption involves various pro-inflammatory cytokines, particularly interleukin-1β (IL-1β), tumor necrosis factor-α (TNF-α), and interleukin-6 (IL-6) [19,20]. These cytokines not only promote local inflammation but also contribute to systemic metabolic dysfunction [21]. Additionally, oxidative stress plays a significant role in HFD-induced intestinal damage, with alterations in antioxidant enzymes such as superoxide dismutase (SOD), catalase, and glutathione peroxidase (GPx) potentially contributing to tissue injury and cellular dysfunction [22,23]. Furthermore, HFD-induced obesity has been associated with increased apoptosis in various tissues, including the intestine [24,25]. The regulation of apoptotic pathways, involving both pro-apoptotic (Bax) and anti-apoptotic (Bcl-2, Bcl-xL) proteins, as well as various caspases, is crucial for maintaining intestinal tissue homeostasis [26]. Disruption of these pathways can lead to excessive cell death, compromised barrier function, and enhanced inflammatory responses [11,27].

One of the proposed treatment options for the harmful effects of obesity is the application of functional foods that have beneficial influences on health beyond their nutritional value. In recent years, there has been growing interest in developing natural compounds and functional foods that could potentially mitigate HFD-induced intestinal damage and inflammation [28,29]. A growing body of evidence suggests that natural bioactive compounds can counteract the adverse effects of HFD by modulating inflammation, oxidative stress, and apoptosis [30,31]. Among these, insect-derived proteins have gained increasing attention due to their high nutritional value and potential health benefits [32,33]. Mealworms (*Tenebrio molitor*) are a rich source of protein, essential amino acids, and bioactive peptides, which have been reported to exhibit antioxidant, anti-inflammatory, and anti-obesity properties [34,35]. Hydrolyzed proteins from mealworms have demonstrated bioactivity in various biological systems, making them promising candidates for therapeutic interventions against metabolic disorders.

Recent studies have highlighted the beneficial effects of bioactive peptides derived from enzymatic hydrolysis of mealworm protein [36]. These peptides have been shown to regulate oxidative stress by enhancing antioxidant enzyme activity, such as SOD, catalase, and GPx, thereby reducing the levels of reactive oxygen species (ROS) [37]. Protaetia Brevitarsis-derived protein hydrolysate (PBPH), a novel peptide extraction derived from mealworms, has shown promise in preliminary studies for its potential anti-inflammatory and anti-oxidative properties. However, its effects on HFD-induced intestinal inflammation and the underlying mechanisms remain largely unexplored. Understanding how PBPH influences adipokine expression, inflammatory responses, oxidative stress, and apoptotic pathways in the context of HFD-induced intestinal damage could provide valuable insights for developing therapeutic strategies against obesity-related complications [38].

This study aimed to investigate the effects of PBPH extraction from mealworm on inflammation, oxidative stress, and apoptosis in an HFD-induced obese mouse model. Specifically, the expression levels of adipokines, pro-inflammatory cytokines, antioxidant enzymes, and apoptotic markers in intestinal tissues were evaluated. By elucidating the potential protective effects of PBPH, we gained new insights into the role of insect-derived bioactive peptides in managing metabolic dysfunction and gut health. These findings could provide valuable insights into the potential therapeutic applications of insect-derived peptides in treating obesity-related intestinal inflammation and dysfunction.

## 2. Results

### 2.1. The Effect of PBPH on Obesity and Serum Lipid Markers

To study the effect of PBPH in colon from HFD animal model, we proceeded with the experiment as visualized in Figure 1A. The timeline illustrates the progression of HFD over time and its impact on mouse morphology and visceral anatomy (Figure 1B). Significant visceral fat accumulation and overall increased body mass were observed in pre-treatment images. PBPH treatment images highlight the efficacy of PBPH in reducing visceral adiposity. As supportive data, we measured the body weight changes over an 11-week period (Figure 1C). The control group shows a steady, moderate increase in body weight, reflecting normal growth without dietary manipulation. The HFD group exhibits a more rapid and higher trajectory of weight gain, indicative of the obesogenic effect of the high-fat diet. In contrast, the PBPH-treated group shows a trajectory of weight gain that, while initially similar to the HFD group, becomes attenuated over time. This suggests that PBPH treatment effectively slows down the rate of weight gain induced by the high-fat diet. Next, we analyzed the blood lipid markers including total cholesterol, triglycerides, low-density lipoprotein (LDL), and high-density lipoprotein (HDL) (Figure 1D). The results show a marked increase in total cholesterol, triglycerides, and LDL levels in mice on the HFD compared to controls, which were significantly reduced by the PBPH treatment. Conversely, HDL levels, which decreased under HFD, were partially restored with the intervention. These findings suggest that the PBPH treatment effectively ameliorates dyslipidemia induced by HFD.

### 2.2. The Effect of PBPH on Adipokine Level and Antioxidant Enzyme Activity

In adipokine analysis, all analyzed adipokines (leptin, adipsin, resistin, and adiponectin) showed elevated levels in the HFD group and suppressed expression in the HFD-with-PBPH treated group (Figure 2A). The values of leptin, adipsin, resistin, and adiponectin are listed in Table 1. As PBPH showed effectiveness in adipokines, we analyzed antioxidant effects by analyzing SOD, catalase levels, and GPx activity (Figure 2B). SOD and catalase levels were increased in the HFD group and decreased in the HFD-with-PBPH treated group. GPx activities were suppressed in the HFD group and increased in the HFD-with-PBPH treated group. Treatment with PBPH restores these antioxidant markers towards baseline levels. The values of SOD, catalase, and GPx activities are listed in Table 2. These findings suggest that PBPH not only mitigates lipid-induced inflammation but also enhances cellular antioxidant defenses, thereby potentially reducing oxidative stress linked to dietary fats. The dual action of PBPH underscores its therapeutic potential in managing diet-induced metabolic disturbances and oxidative damage.

### 2.3. The Effect of PBPH on Inflammation and Its Mechanism

We confirmed the anti-inflammatory effect of PBPH by analyzing mRNA expression levels of inflammation-related genes including IL-1β, TNF-α, and IL-6 (Figure 3A). In the HFD group, there is a notable upregulation of these cytokines, reflecting the pro-inflammatory impact of the diet. The levels of IL-1β, TNF-α, and IL-6 are significantly elevated in the HFD group, underscoring the diet’s capacity to induce systemic inflammation. However, upon administration of PBPH, these cytokines exhibit a marked decrease, suggesting that PBPH effectively counters the inflammatory response triggered by HFD. The results highlight PBPH’s potential as an anti-inflammatory agent, capable of significantly reducing key cytokines associated with inflammation. To explore the underlying mechanisms of anti-inflammatory activity of PBPH, we analyzed inflammation process-related genes (*Toll-Like Receptor 4*, *TLR4*; *Mitogen-Activated Protein Kinase*, *MAPK*; *Extracellular Signal-Regulated Kinase*, *ERK*; *c-Jun N-terminal Kinase*, *JNK*; *p38*; and *Activating Protein-1*, *AP-1*) (Figure 3B). The results demonstrated a significant upregulation of these markers in mice subjected to an HFD, indicating enhanced inflammatory signaling. Notably, upon treatment with PBPH, there was a marked decrease in the expression levels of these molecules across all examined pathways. This reduction suggests that PBPH effectively counters the pro-inflammatory effects induced by an HFD by downregulating the expression of key signaling molecules (ERK, JNK, p38) involved in TLR4–MAPK–AP-1 pathways. To analyze the histological changes in intestine tissue, we proceeded HE staining to examine basic tissue morphology and to grade inflammation score. Under light microscopy, we observed loss of goblet cells and the structure of crypts in the HFD group (Figure 3C). Inflammatory scores were graded by pathologists by the following standard; 0: no inflammation, 1: slight chronic inflammation, 2: mild chronic inflammation, 3: mild inflammation, 4: severe inflammation. In the HFD group, inflammation score was increased and PBPH treatment reduced the inflammation score. Under a fluorescence microscope, we examined the expression of TLR4 (Figure 3D). As we expected, the HFD group showed elevated expressions of TLR4 and AP-1 and PBPH treatment suppressed the elevation in these inflammation-related markers.

### 2.4. The Effect of PBPH on Apoptosis

We analyzed the change in apoptotic gene (*B-cell lymphoma 2*, *Bcl-2*; *B-cell lymphoma-extra large*, *Bcl-xL*; *BCL2-associated X protein*, *Bax*; *caspase 3*; *caspase 8*; and *caspase 9*) expressions (Figure 4A). Anti-apoptotic genes (*Bcl-2* and *Bcl-xL*) were suppressed, and pro-apoptotic genes (*Bax*) were elevated in the HFD group, and PBPH treatment recovered the apoptotic gene changes. For apoptosis cascade components, we selected *caspase 3*, *caspase 8*, and *caspase 9* and examined the expression change. As we expected, *caspase 3*, *caspase 8*, and *caspase 9* levels were elevated in the HFD group and decreased with PBPH treatment. Next, we checked the change in apoptotic proteins (Fas cell surface death receptor, FAS and caspase 8) in intestine tissue by IHC staining. In the HFD group, FAS and caspase 8 expressions were elevated, while PBPH suppressed the expression (Figure 4B). These findings collectively suggest that PBPH may offer protective effects against diet-induced cellular apoptosis by modulating both intrinsic and extrinsic apoptotic pathways.

## 3. Discussion

Obesity and its related metabolic disorders have become a major public health concern worldwide, largely due to changes in dietary habits and sedentary lifestyles. HFDs are known to induce metabolic disturbances by promoting oxidative stress, chronic inflammation, and apoptosis in various tissues, including the gastrointestinal tract [39,40]. In this study, the effects of mealworm-derived PBPH extraction on HFD-induced intestinal inflammation, oxidative stress, and apoptosis in mice were examined. The results demonstrate that PBPH supplementation significantly mitigates the adverse effects of HFD by regulating adipokine secretion, modulating oxidative stress-related enzymes, and influencing apoptotic pathways.

### 3.1. Regulation of Adipokine Secretion and Inflammatory Response

Adipokines, including leptin, adipsin, resistin, and adiponectin, play a crucial role in metabolic homeostasis and inflammation [16]. Leptin, for instance, is known to promote pro-inflammatory responses by activating macrophages and increasing cytokine production, whereas adiponectin exhibits anti-inflammatory and insulin-sensitizing properties [41]. These findings indicate that an HFD significantly increases leptin, adipsin, and resistin levels, while PBPH supplementation effectively suppresses these elevations. This suggests that PBPH possesses the potential to counteract HFD-induced adipokine imbalance, thereby mitigating metabolic inflammation. The dysregulation of adipokine production and secretion is a hallmark of obesity-related disorders [42]. The results showed significantly elevated levels of leptin, adipsin, and resistin in the intestinal tissue of HFD-fed mice, while adiponectin showed a non-significant increase. These findings align with previous studies demonstrating that HFD consumption leads to altered adipokine profiles [43]. The marked elevation in leptin levels (from 48.60 ± 27.39 to 8623.55 ± 1779.44 pg/mL) in the HFD group suggests the development of leptin resistance, a common feature in obesity [44]. Importantly, PBPH treatment substantially reduced these elevated adipokine levels, particularly leptin (reduced to 2179 ± 1758.40 pg/mL), indicating its potential to restore adipokine balance.

Furthermore, a significant upregulation of pro-inflammatory cytokines, including IL-1β, TNF-α, and IL-6, in the intestinal tissues of HFD-fed mice was observed [13]. These cytokines are key mediators of chronic low-grade inflammation, which is commonly associated with obesity-related metabolic disorders [45]. The administration of PBPH significantly reduced the expression of these inflammatory markers, indicating its anti-inflammatory effects. Previous studies have reported that bioactive peptides derived from enzymatic hydrolysis of insect proteins can suppress inflammatory responses by modulating the NF-κB and MAPK signaling pathways [46]. The results of this study are consistent with these findings, suggesting that PBPH may exert its anti-inflammatory effects through similar molecular mechanisms. This modulation of inflammatory mediators is particularly important as chronic inflammation in the intestine can lead to barrier dysfunction and increased susceptibility to various gastrointestinal disorders [47]. The mechanism by which PBPH attenuates inflammatory responses may involve multiple pathways, including the regulation of NF-κB signaling, though further investigation is needed to fully elucidate these mechanisms.

### 3.2. Modulation of Oxidative Stress

Oxidative stress is a key contributor to the pathogenesis of obesity and related metabolic dysfunctions [22]. It results from an imbalance between ROS production and antioxidant defense mechanisms, leading to cellular damage and dysfunction [48]. In this study, the levels of key antioxidant enzymes, including SOD, catalase, and GPx, in intestinal tissues were examined. The results show that HFD consumption leads to a significant increase in SOD and catalase levels while reducing GPx activity, suggesting an adaptive response to excessive oxidative stress [49].

Interestingly, PBPH supplementation prevented the HFD-induced elevation in SOD and catalase levels while restoring GPx activity. This finding indicates that PBPH may help maintain redox homeostasis by enhancing antioxidant defense mechanisms [50]. The anti-oxidative properties of mealworm-derived bioactive peptides may be attributed to their ability to scavenge free radicals and upregulate the expression of endogenous antioxidant enzymes [51]. Studies have shown that certain insect-derived peptides contain amino acid sequences with strong radical-scavenging activities, which may explain the observed protective effects of PBPH against oxidative stress [52,53].

### 3.3. Influence on Apoptotic Pathways

Apoptosis, or programmed cell death, is a tightly regulated process that plays a crucial role in maintaining cellular homeostasis [54]. However, excessive apoptosis can contribute to tissue damage and dysfunction, particularly in metabolic disorders such as obesity [55]. In this study, the expression levels of key apoptotic markers, including *Bax*, *Bcl-2*, *Bcl-xL*, and *caspases 3*, *8*, and *9*, in intestinal tissues were examined.

The results indicate that HFD consumption significantly upregulated *Bax*, *caspase-8*, and *caspase-9*, whereas *caspase-3* showed a non-significant upward trend, while downregulating anti-apoptotic genes (Bcl-2 and Bcl-xL) [56]. This suggests that HFD induces apoptosis in intestinal tissues, potentially contributing to gut barrier dysfunction and increased intestinal permeability [57]. PBPH supplementation effectively reversed these changes, reducing pro-apoptotic gene expression while restoring anti-apoptotic gene levels. These findings suggest that PBPH may exert protective effects against HFD-induced apoptosis by modulating intrinsic and extrinsic apoptotic pathways. The modulation of both the intrinsic (involving caspase 9) and extrinsic (involving caspase 8) apoptotic pathways suggests that PBPH has broad-spectrum anti-apoptotic properties.

The mechanisms underlying the anti-apoptotic effects of PBPH remain to be fully elucidated; however, previous studies have suggested that bioactive peptides can regulate apoptosis by modulating mitochondrial function and inhibiting caspase activation [58]. It is possible that PBPH-derived peptides interact with key apoptotic regulators to suppress excessive cell death, thereby preserving intestinal integrity and function.

### 3.4. Potential Implications and Future Directions

The histological findings provide compelling evidence for the protective effects of PBPH against HFD-induced intestinal injury. The loss of goblet cells and disruption of crypt structure observed in the HFD group represents significant structural damage that could compromise intestinal barrier function. The reduction in inflammation scores following PBPH treatment, coupled with the preservation of tissue architecture, suggest that PBPH can maintain intestinal integrity under HFD conditions. This is further supported by the immunohistochemical analysis showing reduced expression of FAS and caspase 8 in PBPH-treated tissues.

The bioactive properties of PBPH may be attributed to its unique peptide composition derived from mealworms. Insect-derived peptides have been shown to possess various biological activities, including anti-inflammatory, antioxidant, and immunomodulatory properties. The size-selective isolation process used in our PBPH preparation (<3 kDa peptides) may have enriched for particularly bioactive peptide fragments. Future studies should focus on identifying the specific peptide sequences responsible for these beneficial effects and elucidating their structure–function relationships.

The potential therapeutic applications of PBPH extend beyond its local effects in the intestine. Given the crucial role of intestinal health in systemic metabolism and inflammation, the ability of PBPH to protect against HFD-induced intestinal injury could have broader implications for metabolic health. Additionally, the use of insect-derived peptides represents a sustainable approach to developing therapeutic agents, as insect farming has a significantly lower environmental impact compared to traditional livestock farming.

The findings of this study provide compelling evidence that PBPH supplementation can mitigate HFD-induced metabolic disturbances by regulating adipokine secretion, reducing inflammation, modulating oxidative stress, and suppressing apoptosis. These results highlight the potential of mealworm-derived bioactive peptides as functional ingredients for managing obesity-related disorders. Given the increasing interest in alternative protein sources and sustainable nutrition, insect-derived proteins and peptides may offer a promising approach for developing functional foods and nutraceuticals targeting metabolic health.

However, several questions remain unanswered. First, the specific bioactive peptides responsible for the observed effects of PBPH need to be identified. Future studies should focus on characterizing the peptide sequences present in PBPH and investigating their individual bioactivities. Additionally, while our study focused on intestinal tissues, it would be valuable to explore the systemic effects of PBPH on other metabolic organs, such as the liver, adipose tissue, and skeletal muscle.

Several limitations of our study should be acknowledged. First, while we demonstrated significant protective effects of PBPH, the optimal dosing regimen needs to be established. Second, this study did not include mass spectrometry-based peptide sequencing, and PBPH, being a peptide hydrolysate, represents a mixture of bioactive peptides rather than a single purified component. While our primary focus was on the overall biological activities of PBPH, future studies will aim to identify and characterize the specific peptide(s) responsible for the observed effects through detailed profiling and functional validation. Third, the quantification of total polyphenols in PBPH was not performed as part of this work. Although this was beyond the scope of the current study, we acknowledge that such information could further strengthen the interpretation of our results. In future studies, we plan to include detailed phytochemical profiling, including polyphenol quantification, to elucidate their potential contribution to the observed effects. Fourth, the exact molecular mechanisms underlying PBPH’s effects on different signaling pathways require further investigation. Future studies should employ molecular approaches to identify specific cellular targets and signaling cascades affected by PBPH treatment. Moreover, further research is needed to understand the long-term safety and efficacy of PBPH supplementation. Although insect-derived proteins are generally considered safe for human consumption, clinical studies are required to confirm their potential benefits in human populations.

Nevertheless, the translational potential of the findings of this study is particularly exciting. The development of PBPH as a therapeutic agent could provide a novel approach to preventing and treating obesity-related intestinal inflammation and dysfunction. Moreover, the use of insect-derived peptides aligns with growing interest in sustainable and alternative protein sources for both nutritional and therapeutic applications.

## 4. Materials and Methods

### 4.1. Animal Model

Female ICR mice (6-week-old) were purchased from Orient Bio Co. (Sungnam, Republic of Korea). Female ICR mice were selected because they are commonly used in metabolic and nutritional studies, and they exhibit consistent susceptibility to diet-induced metabolic changes. Female rodents provide more stable and reproducible outcomes, with less variability in weight gain and metabolic responses compared to males. A total of 18 mice were randomly divided into three groups (*n* = 6 for each group) for subsequent experimental procedures. Animals were housed in a controlled environment (22 ± 2 °C, 55 ± 5% humidity, 12 h light/dark cycle) with unlimited access to food and water. The in vivo experiments were approved and conducted in accordance with the guidelines by the Yeungnam University Institutional Animal Care and Use Committee (approval number: YUMC-AEC2021-021). All mice were randomly divided into three groups and received the following treatment:(1)N (normal): Normal control mice were fed with a standard chow diet (2.93 kcal/g, approximately 10% kcal from fat; Orient Bio, Seongnam, Republic of Korea).(2)Fat: Mice were fed with a high-fat diet (60% of total calories derived from fat, remaining 40% from carbohydrates and protein, energy density 5.24 kcal/g; D12492, The Jackson Laboratory, Sacramento, CA, USA).(3)PBPH: Mice were fed with a high-fat diet and treated with PBPH extraction via the gastric gavage route (16 mg/100 g of body weight/daily) for 8 weeks.

PBPH was dissolved in sterile phosphate-buffered saline (PBS). Anesthesia was induced using isoflurane through a controlled inhalation method and after collecting animal samples, all animals were sacrificed by cervical dislocation.

### 4.2. Preparation of PBPH Extraction

PBPH extraction was prepared from mealworm (Yeochoun Bugs Land, Yecheon, Republic of Korea). Four percent (*w*/*v*) suspension of mealworm powder in distilled water was heated at 90 °C for 20 min, and alcalase was used for hydrolysis. The ratio of enzyme to mealworm was 1:100, *w*/*v*. After hydrolysis, the hydrolysates were subjected to centrifugation at 13,000× *g* for 20 min to collect the supernatant. The supernatant was then processed through sequential ultrafiltration steps. Initially, centrifugal filter devices with a molecular weight cut-off (MWCO) of 3 kDa (Amicon Ultra-15, EMD Millipore Corp., Burlington, MA, USA) were used at 5000× *g* for 2 h. Subsequently, an additional ultrafiltration step was performed using centrifugal filter devices with an MWCO of 1 kDa (Macrosep Advance, Pall Corp., Port Washington, NY, USA). The protein hydrolysates with <1 kDa and 1–3 kDa (<3 kDa) were lyophilized and stored at −80 °C for further analyses.

### 4.3. Real-Time Quantitative PCR

Collected intestine samples were kept in a −80 °C deep freezer till the RNA extraction. Maxwell^TM^ 16 instrument (Promega Corporation, Madison, WI, USA) was used with Maxwell^®^ RSC simply RNA cell kit to extract tissue RNA. Using GoScript™ Reverse Transcriptase (Promega Corporation, Madison, WI, USA), 1 μg of RNA was used to synthesize cDNA. Real-time quantitative PCR was performed by StepOnePlus™ Real-Time PCR System (Applied Biosystems^®^ Inc., Foster City, CA, USA) with Luna Universal qPCR Master Mix (NEB, Boston, MA, USA). The sequences of used primers are listed in Table 3.

### 4.4. Tissue Sample Analysis

Collected colon tissue samples were analyzed by the provided instruction from ELISA kits. The tissue levels of SOD, GPX, and catalase were measured by ab285309 (Abcam, Cambridge, MA, USA), MBS456700 (MyBioSource, San Diego, CA, USA), and 707002 (Cayman Chemical Company, MI, USA), respectively.

### 4.5. Histological Analysis

Collected tissue samples were fixed in 4% formalin solution and processed by dehydration, clearing, and paraffin embedding. Formed paraffin blocks were cut into 4 um thickness and applied on glass slides. After deparaffinization and hydration, hematoxylin and eosin staining was performed for basic information of microanatomy of tissue. For immunohistochemistry (IHC) staining, antigen retrieval and blocking steps were performed by citrate buffer and 5% bovine serum albumin (BSA) solution. Primary antibodies were applied at a dilution of 1:100 and incubated overnight at 4 °C. The following primary antibodies were used: TLR4 (19811-1-AP, Thermo, Waltham, MA, USA), AP-1 (NBP1-89544, Novus, Centennial, CO, USA), FAS (MA5-14882, Thermo), and Caspase-8 (NB100-56116, Novus, Centennial, CO, USA). After washing, secondary antibodies were applied for 1 h at room temperature: Alexa Fluor 594–conjugated goat anti-rabbit IgG (1:1000; Abcam, Cambridge, MA, USA) and Alexa Fluor 594–conjugated goat anti-mouse IgG (1:1000; Abcam, Cambridge, MA, USA). Finally, slides were mounted using DAPI-containing mounting medium (H-1200, Vector Laboratories, Burlingame, CA, USA). Slides were examined under a light microscope and a fluorescent microscope.

### 4.6. Inflammation Scoring

Inflammation was evaluated using a semi-quantitative grading system under high-power field (HPF, ×400) examination. The grading criteria were as follows: Grade 0, no inflammatory cells; Grade 1, mild inflammation with fewer than 25 inflammatory cells/HPF; Grade 2, moderate inflammation with 25–124 inflammatory cells/HPF; and Grade 3, severe inflammation with 125 or more inflammatory cells/HPF.

### 4.7. Statistical Analysis

Data are expressed as the mean ± standard deviation (SD). Statistically significant differences were determined by a *t*-test and one-way analysis of variance with Tukey’s post hoc test. Differences were considered significant at a *p* value < 0.05, <0.01, and <0.001.

## 5. Conclusions

This study demonstrates that PBPH, a novel mealworm-derived peptide extract, effectively protects against HFD-induced intestinal injury through multiple mechanisms, including modulation of adipokine expression, suppression of inflammation, maintenance of antioxidant defenses, and regulation of apoptotic pathways. These findings provide a strong foundation for further development of PBPH as a therapeutic agent for obesity-related intestinal disorders and highlight the potential of insect-derived peptides in medical applications. Future research should focus on optimizing PBPH formulation, establishing safety profiles, and investigating potential synergistic effects with existing treatments for metabolic disorders. These findings contribute to the growing interest in sustainable protein sources and their application in functional foods or nutraceuticals targeting obesity-related disorders. Understanding the mechanisms by which PBPH modulates metabolic and inflammatory responses may pave the way for the development of novel therapeutic strategies to combat obesity and its associated complications.

## Figures and Tables

**Figure 1 ijms-26-10352-f001:**
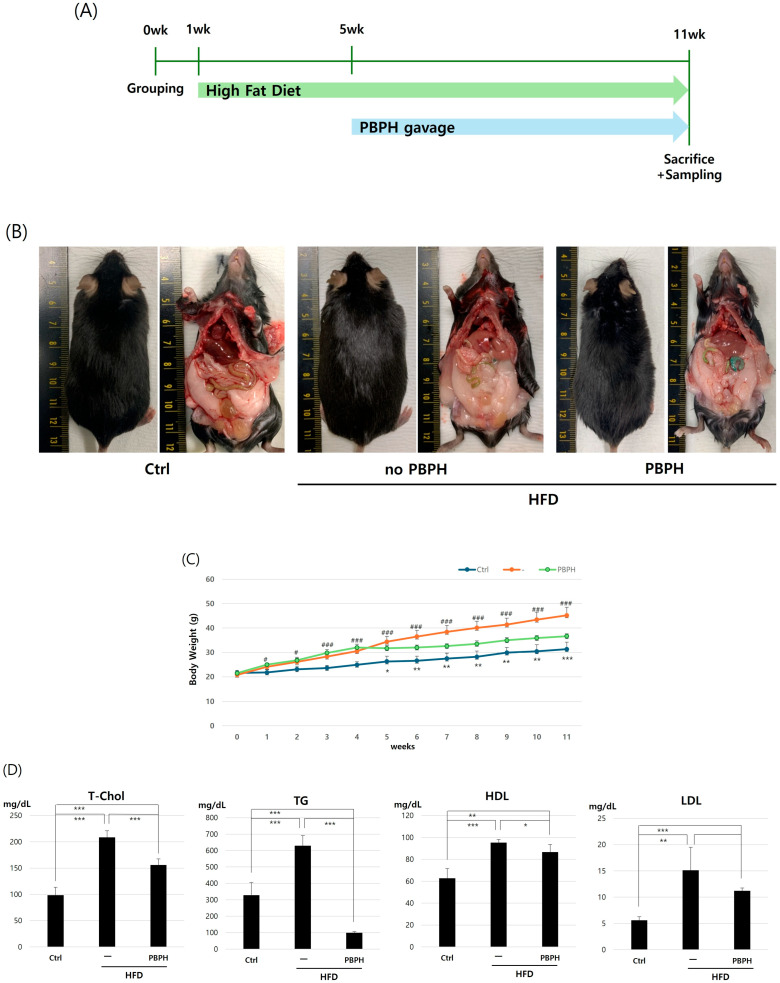
Assessment of PBPH treatment on HFD-induced obesity in mice. (**A**) Experimental timeline. This panel presents the timeline of the experimental setup showing the duration of HFD administration starting from 7 weeks of age, followed by the PBPH gavage. The timeline indicates the specific weeks when the PBPH treatment was initiated and continued alongside the HFD. Ctrl refers to the group fed with a standard chow diet. (**B**) Morphological and visceral anatomy changes. Sequential images from left to right illustrate the morphological changes in mice subjected to an HFD both before and after PBPH gavage. (**C**) Body weight measurements. This graph compares the body weight over an 11-week period. (**D**) Blood analysis. The blood analysis results provide biochemical evidence of PBPH’s impact on metabolic health. Key indicators such as cholesterol, triglycerides, and glucose levels are displayed. All data are presented as mean ± SD (*n* = 6 mice per group). Statistical significance was determined by Student’s *t*-test or one-way ANOVA followed by Tukey’s post hoc test. * *p* < 0.05, ** *p* < 0.01, *** *p* < 0.001, # *p* < 0.05, ### *p* < 0.001.

**Figure 2 ijms-26-10352-f002:**
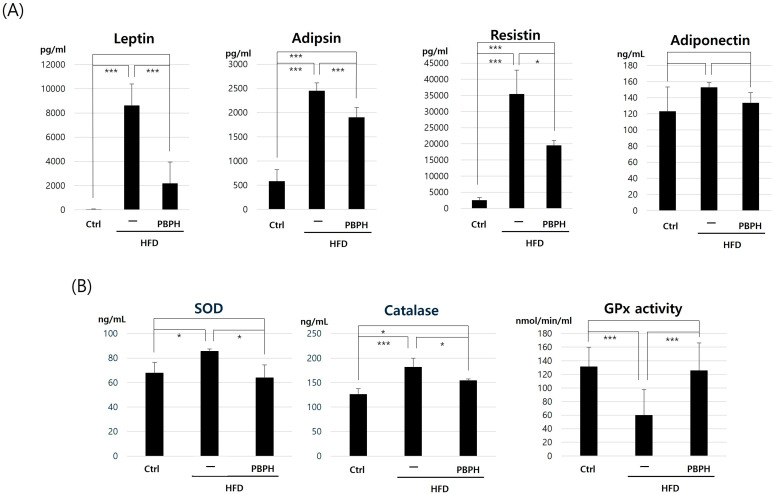
Adipokine levels and antioxidant enzyme activities in colon tissue in response to PBPH treatment. (**A**) Adipokine levels. Displayed are the levels of adipokines across different treatment groups: control (Ctrl), HFD, and HFD treated with PBPH (HFD + PBPH). The plots show changes in several key adipokines such as leptin, resistin, and adiponectin. (**B**) Antioxidant enzyme activities. This panel quantifies the activities of antioxidant enzymes: superoxide dismutase (SOD), catalase and glutathione peroxidase (GPx) activity level. All data are presented as mean ± SD (*n* = 6 mice per group). Statistical significance was determined by Student’s *t*-test or one-way ANOVA followed by Tukey’s post hoc test. * *p* < 0.05, *** *p* < 0.001.

**Figure 3 ijms-26-10352-f003:**
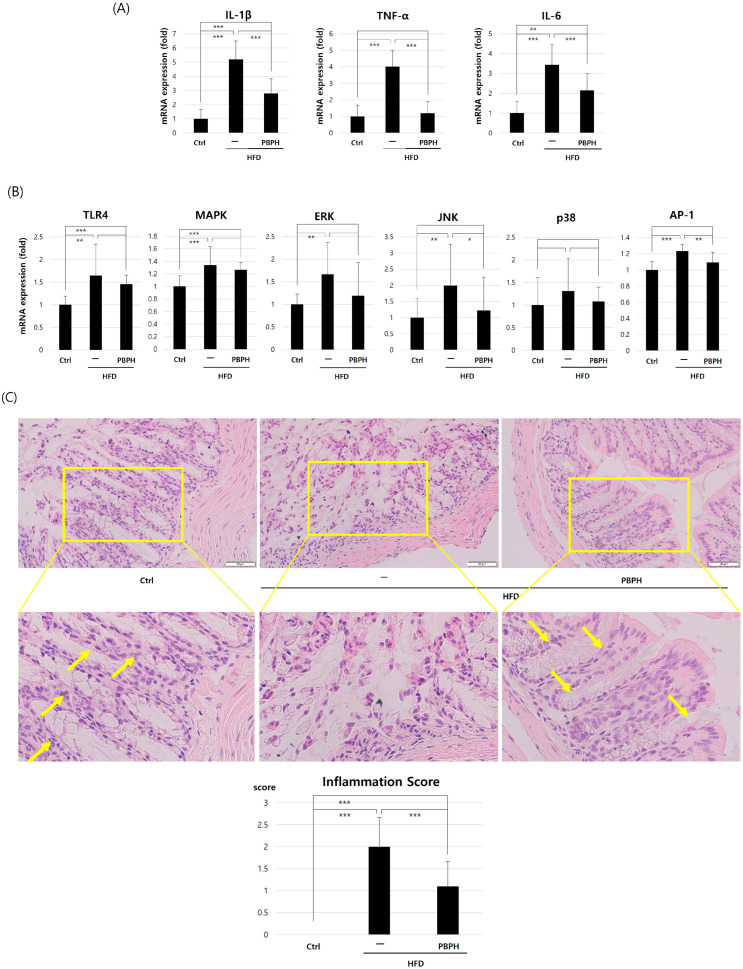
Impact of PBPH on inflammation in colon tissue. (**A**) Inflammatory cytokine levels. This panel illustrates the influence of PBPH treatment on mRNA levels of key inflammatory cytokines such as *TNF-α*, *IL-6*, and *IL-1β*. (**B**) Inflammatory pathway activation. These graphs demonstrate the TLR4-mediated inflammation-related markers. (**C**) Histological evaluation. The representative HE-stained colon tissue displays the level of tissue inflammation. Arrows indicate normal crypt structures, scale bar = 50 μm. (**D**) Immunohistochemistry staining of inflammatory markers (TLR4 and AP-1) in colon tissue. The representative pictures show inflammation-related markers. All data are presented as mean ± SD (*n* = 6 mice per group). Statistical significance was determined by Student’s *t*-test or one-way ANOVA followed by Tukey’s post hoc test. * *p* < 0.05, ** *p* < 0.01, *** *p* < 0.001.

**Figure 4 ijms-26-10352-f004:**
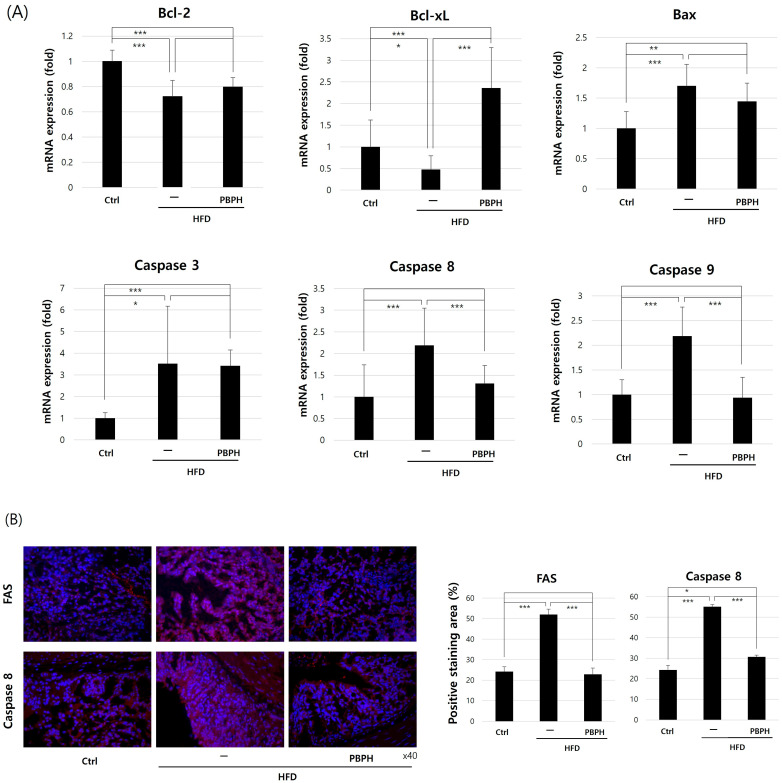
Impact of PBPH on apoptosis in colon tissue. (**A**) mRNA expression levels of apoptosis-related markers. These graphs show the effect of PBPH on apoptosis-related genes. (**B**) Immunohistochemistry staining of apoptotic markers. The representative images show the expression level of FAS and caspase8 in the PBPH-treated HFD model. Magnification = ×40. All data are presented as mean ± SD (*n* = 6 mice per group). Statistical significance was determined by Student’s *t*-test or one-way ANOVA followed by Tukey’s post hoc test. * *p* < 0.05, ** *p* < 0.01, *** *p* < 0.001.

**Table 1 ijms-26-10352-t001:** Colon tissue levels of leptin, adipsin, resistin, and adiponectin.

Adipokine	Ctrl	HFD	HFD + PBPH
Leptin (pg/mL)	48.60 ± 27.39	8623.55 ± 1779.44	2179 ± 1758.40
Adipsin (pg/mL)	581.1 ± 240.93	2450.5 ± 163.03	1902.6 ± 205.02
Resistin (pg/mL)	2568 ± 777.25	35,448 ± 7361.90	19,543.33 ± 1461.24
Adiponectin (ng/mL)	123.12 ± 30.03	152.98 ± 6.03	133.68 ± 12.74

**Table 2 ijms-26-10352-t002:** Colon tissue levels of SOD, catalase, and GPx activity.

Parameter	Ctrl	HFD	HFD + PBPH
SOD (pg/mL)	67.76 ± 8.91	85.04 ± 1.94	63.79 ± 10.76
Catalase (ng/mL)	126.51 ± 11.30	182.14 ± 17.84	154.68 ± 2.94
GPx activity (nmol/min/mL)	131.72 ± 28	59.96 ± 37.84	125.82 ± 40.37

**Table 3 ijms-26-10352-t003:** Primer sequences.

Gene	Forward (5′ to 3′)	Reverse (5′ to 3′)
*AP-1*	GGTTCAGGAAGCCATTGTGGTC	TCAGGCTCATCCAGAGAGTCCA
*Bax*	GATGCGTCCACCAAGAAG	AGTTGAAGTTGCCGTCAG
*Bcl-xL*	GTTCCCTTTCCTTCCATCC	TAGCCAGTCCAGAGGTGAG
*Caspase3*	GGAGTCTGACTGGAAAGCCGAA	CTTCTGGCAAGCCATCTCCTCA
*Caspase8*	ATGGCTACGGTGAAGAACTGCG	TAGTTCACGCCAGTCAGGATGC
*Caspase9*	GCTGTGTCAAGTTTGCCTACCC	CCAGAATGCCATCCAAGGTCTC
*GAPDH*	GTCTCCTCTGACTTCAACAGCG	ACCACCCTGTTGCTGTAGCCAA
*IL-1β*	TGGACCTTCCAGGATGAGGACA	GTTCATCTCGGAGCCTGTAGTG
*IL-6*	TACCACTTCACAAGTCGGAGGC	CTGCAAGTGCATCATCGTTGTTC
*JNK*	GACGCCTTATGTAGTGACTCGC	TCCTGGAAAGAGGATTTTGTGGC
*MAPK*	GCGACTACATTGACCAGCTG	AAGATGCTGCTCAGGTCCTT
*ERK*	ACACCAACCTCTCGTACATCGG	TGGCAGTAGGTCTGGTGCTCAA
*P38*	GAGCGTTACCAGAACCTGTCTC	AGTAACCGCAGTTCTCTGTAGGT
*TLR4*	AGCTTCTCCAATTTTTCAGAACTTC	TGAGAGGTGGTGTAAGCCATGC
*TNF-α*	TTCACTGGAGCCTCGAATGT	ACCTGACCACTCTCCCTTTG

## Data Availability

Data are available from the corresponding author on reasonable request.

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
