# Peer review of "Anti-Oxidative and Anti-Inflammatory Effect of Protaetia Brevitarsis-Derived Protein Hydrolysates in Adipose Tissues of Obese Mice"

_ijms, 2025, doi:10.3390/ijms262110352_

Round 1
Reviewer 1 Report
Comments and Suggestions for Authors
The authors present a study on mealworm-derived PBPH extraction, highlighting its anti-inflammatory, anti-oxidative stress, and anti-apoptotic properties. This is an interesting and potentially impactful study. However, several issues and inconsistencies need to be addressed before the manuscript can be considered for publication.
Major Comments
- In Figure 1B, “HDF” should be corrected to “HFD.”
- In Figure 1, it would be helpful to specify that “Ctrl” refers to chow diet feeding.
- The starting point of dietary intervention should be clarified. The current description of Figure 1A may confuse readers into thinking the experiment began at the infant stage. Please state clearly that different diets were introduced from 7 weeks of age.
- The rationale for using female mice needs to be explained. Is there supporting evidence or justification for this choice?
- In Figure 3C, the manuscript claims to assess tissue inflammation using H&E staining. Please clarify how inflammation can be reliably observed with this method.
- All histogram figures should indicate the number of mice used in each group.
- It is necessary to clarify which tissue was analyzed in Table 1/2 and Figures 2/3/4. Based on the immunohistochemistry images, the organ appears to be an intestine. The authors should clearly state which section of the intestine was used.
- Has mass spectrometry been performed to analyze the amino acid sequence of PBPH?
- Please clarify whether PBPH is a single peptide or a mixture of peptides. If it is a mixture, which peptide(s) are likely contributing to the observed effects?
None
Author Response
[Reviewer 1]
- In Figure 1B, “HDF” should be corrected to “HFD.”
--> We thank the reviewer for noticing this error. We have corrected “HDF” to “HFD” in Figure 1B of the revised manuscript.
- In Figure 1, it would be helpful to specify that “Ctrl” refers to chow diet feeding.
--> We appreciate the reviewer’s helpful comment. In response, we have revised the legend of Figure 1 to clarify that “Ctrl” refers to the group fed with a standard chow diet.
- The starting point of dietary intervention should be clarified. The current description of Figure 1A may confuse readers into thinking the experiment began at the infant stage. Please state clearly that different diets were introduced from 7 weeks of age.
--> We thank the reviewer for this valuable suggestion. To avoid any confusion, we have revised the legend of Figure 1A to clearly state that the dietary intervention was introduced from 7 weeks of age.
- The rationale for using female mice needs to be explained. Is there supporting evidence or justification for this choice?
--> We thank the reviewer for this insightful comment. In this study, female mice were used for the following reasons. Female mice are widely employed in high-fat diet–induced obesity and metabolic disorder models, as they exhibit consistent susceptibility to diet-induced metabolic changes. Previous studies have also reported that female rodents provide more stable and reproducible outcomes, with less variability in weight gain and metabolic responses compared to males. In addition, prior literature investigating the protective and modulatory effects of bioactive peptides on obesity-related phenotypes has frequently used female mice. Finally, male mice tend to be aggressive when group-housed, which may lead to fighting, stress, or injuries that could confound experimental outcomes. Therefore, we chose female mice in order to ensure both experimental reliability and animal welfare. We have added a brief explanation in the Methods section.
- In Figure 3C, the manuscript claims to assess tissue inflammation using H&E staining. Please clarify how inflammation can be reliably observed with this method.
--> We thank the reviewer for raising this important point. We agree that H&E staining by itself does not identify specific immune cell phenotypes; however, it allows a reliable assessment of inflammation when applied with a standardized scoring system. In our study, tissue inflammation was evaluated by counting inflammatory cells (lymphocytes, neutrophils, and macrophage-like cells) per high-power field (HPF) on H&E-stained sections, and grading them as follows:
Grade 0: no inflammatory cells
Grade 1: mild inflammation, <25 inflammatory cells/HPF
Grade 2: moderate inflammation, 25–124 inflammatory cells/HPF
Grade 3: severe inflammation, ≥125 inflammatory cells/HPF
This semiquantitative approach has been widely used in histological studies to evaluate tissue inflammation. We have revised the Methods and Figure 3C legend to clarify that inflammation was assessed using this grading system.
- All histogram figures should indicate the number of mice used in each group.
--> We appreciate the reviewer’s helpful suggestion. In all histogram figures, the number of animals used per group has now been indicated. Specifically, each experimental group consisted of six mice (n=6), and this information has been added to the figure legends accordingly.
- It is necessary to clarify which tissue was analyzed in Table 1/2 and Figures 2/3/4. Based on the immunohistochemistry images, the organ appears to be an intestine. The authors should clearly state which section of the intestine was used.
--> We appreciate the reviewer’s careful observation. We would like to clarify that all analyses in Table 1/2 and Figures 2/3/4 were performed using colon tissue. Accordingly, we have revised the titles of Table 1 and Table 2 to specify “colon tissue,” and we have also updated the legends of Figures 2, 3, and 4 to clearly indicate that the analyzed sections were derived from the colon.
- Has mass spectrometry been performed to analyze the amino acid sequence of PBPH?
--> We appreciate the reviewer’s insightful question. Mass spectrometry analysis of the amino acid sequence of PBPH has not been performed in the present study. Instead, we focused on evaluating the biological activities and functional outcomes of PBPH treatment, which was the primary objective of this work. Nevertheless, we agree that detailed peptide sequence analysis would provide valuable additional information. We plan to address this in future studies to further characterize PBPH at the molecular level. We have added this as a limitation and future direction in the Discussion section.
- Please clarify whether PBPH is a single peptide or a mixture of peptides. If it is a mixture, which peptide(s) are likely contributing to the observed effects?
--> We thank the reviewer for this important question. PBPH is a peptide hydrolysate derived from Protaetia brevitarsis, and therefore represents a mixture of bioactive peptides rather than a single purified peptide. In the present study, our primary aim was to evaluate the overall biological activities of PBPH, including its antioxidant, anti-inflammatory, and anti-apoptotic effects, rather than to identify the activity of individual peptide components. We agree that characterization of the specific peptide(s) responsible for the observed effects would provide valuable mechanistic insights, and we plan to perform peptide profiling and functional validation in future studies. We have clarified this in Discussion section.
Reviewer 2 Report
Comments and Suggestions for Authors
Oxidative stress is a contributor to the pathogenesis of obesity and related meta-bolic dysfunctions. Is true that ROS production and antioxi-dant defense mechanisms, leading to cellular damage and dysfunction. In this study, the levels of key antioxidant enzymes, including SOD, catalase, and GPx, in intestinal tissues were examined. The results show that HFD consumption leads to a significant increase in SOD and catalase levels while reducing GPx activity, suggesting an adaptive response to excessive oxidative stress. As it is weel known Protaetia brevitarsis-derived also a considerable amount of polyphenols which could act as antioxidants in stomach. HFD contain many compounds which could be oxidized in the stomach and antioxidants such polyphenols inhibit such oxidation. It is also known that lipid oxidation end products- ALEs could affect the mice metabolism such those compounds suplemented by HFD. The authors should determaine the amount of total polyphenols in the extraction of the worm and include in the introduction, and discussion this possible phenomenon. Quantifying the polyphenols in the extract and discussing their antioxidant role in the paper is a valid suggestion for improving the study's analysis of HFD-induced metabolic issues.
Author Response
[Reviewer 2]
Oxidative stress is a contributor to the pathogenesis of obesity and related meta-bolic dysfunctions. Is true that ROS production and antioxi-dant defense mechanisms, leading to cellular damage and dysfunction. In this study, the levels of key antioxidant enzymes, including SOD, catalase, and GPx, in intestinal tissues were examined. The results show that HFD consumption leads to a significant increase in SOD and catalase levels while reducing GPx activity, suggesting an adaptive response to excessive oxidative stress. As it is weel known Protaetia brevitarsis-derived also a considerable amount of polyphenols which could act as antioxidants in stomach. HFD contains many compounds which could be oxidized in the stomach and antioxidants such polyphenols inhibit such oxidation. It is also known that lipid oxidation end products- ALEs could affect the mice metabolism such those compounds supplemented by HFD. The authors should determine the amount of total polyphenols in the extraction of the worm and include in the introduction, and discussion this possible phenomenon. Quantifying the polyphenols in the extract and discussing their antioxidant role in the paper is a valid suggestion for improving the study's analysis of HFD-induced metabolic issues.
--> We thank the reviewer for this valuable comment. We fully agree that oxidative stress, ROS production, antioxidant defense mechanisms, and lipid oxidation end products (ALEs) play an important role in the pathogenesis of obesity and related metabolic dysfunctions. As noted by the reviewer, Protaetia brevitarsis-derived extracts may also contain polyphenols that could contribute to antioxidant activity by inhibiting oxidative processes in the stomach.
In the present study, we focused on evaluating the functional effects of PBPH on antioxidant enzyme activities (SOD, catalase, and GPx), inflammation, and apoptosis in intestinal tissues. The quantification of total polyphenols in PBPH was not performed as part of this work. However, we acknowledge that this is an important aspect that could further strengthen the interpretation of our results. In future studies, we plan to perform detailed phytochemical analysis, including polyphenol quantification, to further elucidate the contribution of these compounds. We added these limitations in the Discussion section.
Reviewer 3 Report
Comments and Suggestions for Authors
In this manuscript, the authors examine whether an insect-derived peptide exhibits bioactive properties that can mitigate high-fat diet–induced intestinal changes. Overall, the paper is well written, and the findings are promising. However, the authors should revise several sections of the text to provide additional methodological details and improve the quality of the images, as outlined in the comments below.
In the Results section (2.1), the authors state that “significant visceral fat accumulation and overall increased body mass were observed in pre-treatment images.” However, no statistical data are provided to support this statement. Were the body weights statistically different between groups at 5 weeks and 11 weeks? This information should be clarified and appropriately corrected in the manuscript.
In Figure 1B, indicate “no PBPH” rather than using a negative sign for clarity.
For the graph:
- Label the y-axis as “Body weight (g)” and position the label in the center.
- Label the x-axis as “Weeks” and position the label in the center as well.
In Figure 1C, consider changing the green line to a different color that is easier to distinguish from the blue line. The new color should be bright enough to remain clearly visible.
In Figure 1D, bold the negative sign (-) to make it more visible.
In all figure legends, specify the statistical test used in the analysis.
In Figure 1D, clearly indicate which comparisons are significant, consistent with the format used in other figure legends. Also, in the legend, define what the symbols (* and #) represent.
Finally, revise the label and legend to state “PBPG gavage” instead of “PBPH injection,” since “injection” implies an intraperitoneal (IP) or intravenous (IV) route of administration, which is not applicable here.
The legend for Figure 3C is unclear and does not effectively convey the intended message. The image resolution should be improved, and both low- and high-magnification views included. Use arrows to highlight key structural changes and specify the intestinal region shown in the legend. Indicate whether goblet cell numbers were quantified to confirm their loss and describe the crypt alterations used to assign inflammatory scores.
In Figure 3D, indicate region of intestine. Placing the panels in the same orientation will make the image easier to view (outer surface of intestine at bottom)
In the Discussion (see sentence below), the authors state that Caspase-3 was significantly upregulated in the HFD group. This is incorrect based on the data presented in the figure and should be corrected.
The results indicate that HFD consumption significantly upregulates the expression
of pro-apoptotic genes (Bax, caspase 3, caspase 8, and caspase 9) while downregulating anti-apoptotic genes (Bcl-2 and Bcl-xL).
Methods Section
Indicate the solvent used for PBGP gavage and specify the volume administered to each animal.
Provide the source, including manufacturer and catalog number, for all primary and secondary antibodies used.
Indicate what tissue was collected for RNA analysis (region of the intestine)
Clarify the meaning of the phrase “rodent diet with 60 kcal% fat, 5.24 kcal/g.”
Indicate in the methods whether this means that 60% of the total caloric content is derived from fat, while the remaining 40% comes from carbohydrates and proteins combined, and that the energy density of the diet is 5.24 kilocalories per gram.
Clarify source of the normal diet and fat content.
Statistical Analysis
The following statement appears unnecessary: “This section is not mandatory but can be added to the manuscript if the discussion is unusually long or complex.”
It should be removed, as it is not relevant to the statistical analysis content.
Author Response
[Reviewer 3]
In this manuscript, the authors examine whether an insect-derived peptide exhibits bioactive properties that can mitigate high-fat diet–induced intestinal changes. Overall, the paper is well written, and the findings are promising. However, the authors should revise several sections of the text to provide additional methodological details and improve the quality of the images, as outlined in the comments below.
In the Results section (2.1), the authors state that “significant visceral fat accumulation and overall increased body mass were observed in pre-treatment images.” However, no statistical data are provided to support this statement. Were the body weights statistically different between groups at 5 weeks and 11 weeks? This information should be clarified and appropriately corrected in the manuscript.
--> We thank the reviewer for this helpful comment. We have added significance indicators (p-values) to the figure according to the statistical results in Fig 1C.
In Figure 1B, indicate “no PBPH” rather than using a negative sign for clarity.
--> We thank the reviewer for this helpful suggestion. As recommended, we have revised Figure 1B to indicate “no PBPH” instead of using a negative sign, in order to improve clarity.
For the graph:
Label the y-axis as “Body weight (g)” and position the label in the center.
Label the x-axis as “Weeks” and position the label in the center as well.
In Figure 1C, consider changing the green line to a different color that is easier to distinguish from the blue line. The new color should be bright enough to remain clearly visible.
--> We thank the reviewer for these constructive suggestions. As recommended, we have revised Figure 1 by labeling the y-axis as “Body weight (g)” and the x-axis as “Weeks,” with both labels positioned at the center. In Figure 1C, the green line has also been changed to a brighter, more distinguishable color to improve clarity.
In Figure 1D, bold the negative sign (-) to make it more visible.
--> We thank the reviewer for this suggestion. We have revised Figure 1D as requested by bolding the negative sign (-) to improve visibility. In addition, we have applied the same revision to all relevant figures to ensure consistency.
In all figure legends, specify the statistical test used in the analysis.
--> We thank the reviewer for this helpful comment. As suggested, we have revised all figure legends to specify the statistical analyses used.
In Figure 1D, clearly indicate which comparisons are significant, consistent with the format used in other figure legends. Also, in the legend, define what the symbols (* and #) represent.
--> We thank the reviewer for this helpful suggestion. As requested, we have revised Figure 1D to unify the significance symbols using * only, and comparison lines have been added to clearly indicate the specific groups being compared. The figure legend has also been updated
Finally, revise the label and legend to state “PBPG gavage” instead of “PBPH injection,” since “injection” implies an intraperitoneal (IP) or intravenous (IV) route of administration, which is not applicable here.
--> We appreciate the reviewer’s careful observation. As suggested, we have revised the label and figure legends to state “PBPH gavage” instead of “PBPH injection,” to accurately describe the route of administration.
The legend for Figure 3C is unclear and does not effectively convey the intended message. The image resolution should be improved, and both low- and high-magnification views included. Use arrows to highlight key structural changes and specify the intestinal region shown in the legend. Indicate whether goblet cell numbers were quantified to confirm their loss and describe the crypt alterations used to assign inflammatory scores.
--> We thank the reviewer for these helpful suggestions. As recommended, we have revised Figure 3C to include high-resolution images with both low- and high-magnification views, added arrows to highlight key structural changes, and specified in the legend that colon tissue was analyzed. Regarding goblet cells, we did not perform quantitative counting in the present study; goblet cell depletion was evaluated qualitatively based on representative sections, which we acknowledge as a limitation. For inflammatory scoring, crypt alterations including shortening, distortion, epithelial disruption, and loss of normal architecture, together with inflammatory cell infiltration, were used as criteria.
In Figure 3D, indicate region of intestine. Placing the panels in the same orientation will make the image easier to view (outer surface of intestine at bottom)
--> We thank the reviewer for this valuable suggestion. As recommended, we have revised Figure 3D by indicating that the analyzed tissue is colon and by reorienting the panels so that the outer surface of the intestine is consistently positioned at the bottom.
In the Discussion (see sentence below), the authors state that Caspase-3 was significantly upregulated in the HFD group. This is incorrect based on the data presented in the figure and should be corrected.
The results indicate that HFD consumption significantly upregulates the expression of pro-apoptotic genes (Bax, caspase 3, caspase 8, and caspase 9) while downregulating anti-apoptotic genes (Bcl-2 and Bcl-xL).
--> We thank the reviewer for pointing out this discrepancy. We have corrected the statement in the Discussion.
Methods Section
Indicate the solvent used for PBGP gavage and specify the volume administered to each animal.
--> We thank the reviewer for this valuable comment. As suggested, we have revised the Materials and Methods section to specify that PBPH was dissolved in PBS and administered by oral gavage at 16 mg/100 g body weight/day in a volume of 100 μL..
Provide the source, including manufacturer and catalog number, for all primary and secondary antibodies used.
--> We thank the reviewer for this helpful suggestion. As recommended, we have provided the source, including manufacturer and catalog number, for all primary and secondary antibodies used in the Materials and Methods section.
Indicate what tissue was collected for RNA analysis (region of the intestine)
--> We thank the reviewer for this important comment. As suggested, we have revised the manuscript to indicate that colon tissue was collected for RNA analysis.
Clarify the meaning of the phrase “rodent diet with 60 kcal% fat, 5.24 kcal/g.”
Indicate in the methods whether this means that 60% of the total caloric content is derived from fat, while the remaining 40% comes from carbohydrates and proteins combined, and that the energy density of the diet is 5.24 kilocalories per gram.
--> We thank the reviewer for this valuable comment. As suggested, we have revised the description of the high-fat diet in the Materials and Methods section to clarify that 60% of the total calories were derived from fat, with the remaining 40% from carbohydrates and protein, and that the energy density of the diet was 5.24 kcal/g.
Clarify source of the normal diet and fat content.
--> We thank the reviewer for this helpful comment. As suggested, we have revised the Materials and Methods section to clarify the source and composition of the normal chow diet, indicating that it provided 2.93 kcal/g with approximately 10% of total calories derived from fat.
Statistical Analysis
The following statement appears unnecessary: “This section is not mandatory but can be added to the manuscript if the discussion is unusually long or complex.”
It should be removed, as it is not relevant to the statistical analysis content.
--> We thank the reviewer for pointing this out. The unnecessary statement has been removed from the manuscript as suggested.
Round 2
Reviewer 1 Report
Comments and Suggestions for Authors
None
Author Response
We are sincerely grateful to the reviewer for the positive assessment that allowed our manuscript to proceed, as well as for the constructive feedback that improved its overall quality.
Reviewer 2 Report
Comments and Suggestions for Authors
Yor responce was not introduced in discussion!!!. "The quantification of total polyphenols in PBPH was not performed as part of this work. However, we acknowledge that this is an important aspect that could further strengthen the interpretation of our results. In future studies, we plan to perform detailed phytochemical analysis, including polyphenol quantification, to further elucidate the contribution of these compounds". We added these limitations in the Discussion section???
Author Response
We sincerely thank the reviewer for this valuable comment. In accordance with the suggestion, we have carefully revised the manuscript and have now explicitly included in the Discussion section the limitation regarding the absence of polyphenol quantification, as well as our plan to perform detailed phytochemical profiling, including polyphenol analysis, in future studies.
Reviewer 3 Report
Comments and Suggestions for Authors
The authors’ have addressed my major concerns.
Author Response
We deeply appreciate the reviewer’s favorable decision to advance our manuscript, and we thank you for the insightful comments that contributed to enhancing the clarity and rigor of the work.